# HyperSAGE: Generalizing Inductive Representation Learning on Hypergraphs

## Abstract

Graphs are the most ubiquitous form of structured data representation used in machine learning. They model, however, only pairwise relations between nodes and are not designed for encoding the higher-order relations found in many real-world datasets. To model such complex relations, hypergraphs have proven to be a natural representation. Learning the node representations in a hypergraph is more complex than in a graph as it involves information propagation at two levels: within every hyperedge and across the hyperedges. Most current approaches first transform a hypergraph structure to a graph for use in existing geometric deep learning algorithms. This transformation leads to information loss, and sub-optimal exploitation of the hypergraph's expressive power. We present HyperSAGE, a novel hypergraph learning framework that uses a two-level neural message passing strategy to accurately and efficiently propagate information through hypergraphs. The flexible design of HyperSAGE facilitates different ways of aggregating neighborhood information. Unlike the majority of related work which is transductive, our approach, inspired by the popular GraphSAGE method, is inductive. Thus, it can also be used on previously unseen nodes, facilitating deployment in problems such as evolving or partially observed hypergraphs. Through extensive experimentation, we show that HyperSAGE outperforms state-of-the-art hypergraph learning methods on representative benchmark datasets. We also demonstrate that the higher expressive power of HyperSAGE makes it more stable in learning node representations as compared to the alternatives.

## 1 Introduction

Graphs are considered the most prevalent structures for discovering useful information within a network, especially because of their capability to combine object-level information with the underlying inter-object relations (Wu et al., 2020). However, most structures encountered in practical applications form groups and relations that cannot be properly represented using pairwise connections alone, hence a graph may fail to capture the collective flow of information across objects. In addition, the underlying data structure might be evolving and only partially observed. Such dynamic higher-order relations occur in various domains, such as social networks (Tan et al., 2011), computational chemistry (Gu et al., 2020), neuroscience (Gu et al., 2017) and visual arts (Arya et al., 2019), among others. These relations can be readily represented with *hypergraphs*, where an edge can connect an arbitrary number of vertices as opposed to just two vertices in graphs. Hypergraphs thus provide a more flexible and natural framework to represent such multi-way relations (Wolf et al., 2016), however, this requires a representation learning technique that exploits the full expressive power of hypergraphs and can generalize on unseen nodes from a partially observed hypergraph.

Recent work in the field of geometric deep learning have presented formulations on graph structured data for the tasks of node classification (Kipf & Welling, 2016), link prediction (Zhang & Chen, 2018), or the classification of graphs (Zhang et al., 2018b). Subsequently, for data containing higher-order relations, a few recent papers have presented hypergraph-based learning approaches on similar tasks (Yadati et al., 2019; Feng et al., 2019). A common implicit premise in these papers is that a hypergraph can be viewed as a specific type of regular graph. Therefore, reduction of hypergraph learning problem to that of a graph should suffice. Strategies to reduce a hypergraph to a graph include transforming the hyperedges into multiple edges using clique expansion (Feng et al., 2019; Jiang et al., 2019; Zhang et al., 2018a), converting to a heterogeneous graph using star

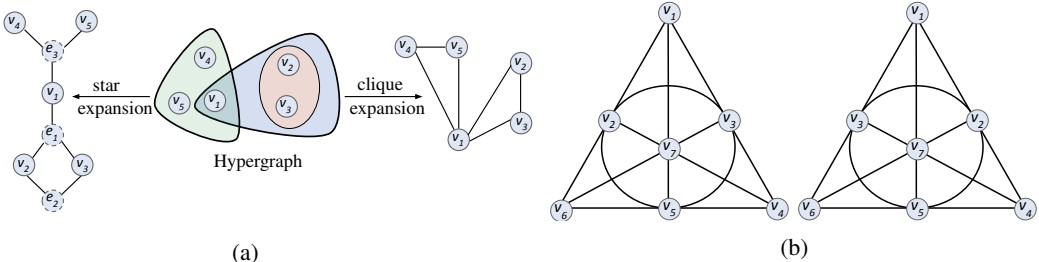

(a)                                                    (b)

Figure 1: (a) Example showing reduction of a hypergraph to a graph using clique and star expansion methods. The clique expansion loses the unique information associated with the hyperedge defined by the set of nodes $\{v_2, v_3\}$, and it cannot distinguish it from the hyperedge defined by the nodes $\{v_1, v_2, v_3\}$. Star expansion creates a heterogeneous graph that is difficult to handle using most well-studied graph methods (Hein et al., 2013). (b) Schematic representations of two Fano planes comprising 7 nodes and 7 hyperedges (6 straight lines and 1 circle.). The second Fano plane is a copy of the first with nodes $v_2$ and $v_3$ permuted. These two hypergraphs cannot be differentiated when transformed to a graph using clique expansion.

expansion (Agarwal et al., 2006), and replacing every hyperedge with an edge created using a certain predefined metric (Yadati et al., 2019). Yet these methods are based on the wrong premise, motivated chiefly by a larger availability of graph-based approaches. By reducing a hypergraph to regular graph, these approaches make existing graph learning algorithms applicable to hypergraphs. However, hypergraphs are not a special case of regular graphs. The opposite is true, regular graphs are simply a specific type of hypergraph (Berge & Minieka, 1976). Therefore, reducing the hypergraph problem to that of a graph cannot fully utilize the information available in hypergraph. Two schematic examples outlining this issue are shown in Fig.1. To address tasks based on complex structured data, a hypergraph-based formulation is needed that complies with the properties of a hypergraph.

A major limitation of the existing hypergraph learning frameworks is their inherently transductive nature. This implies that these methods can only predict characteristics of nodes that were present in the hypergraph at training time, and fail to infer on previously unseen nodes. The transductive nature of existing hypegraph approaches makes them inapplicable in, for example, finding the most promising target audience for a marketing campaign or making movie recommendations with new movies appearing all the time. An inductive solution would pave the way to solve such problems using hypergraphs. The inductive learning framework must be able to identify both the node's local role in the hypergraph, as well as its global position (Hamilton et al., 2017). This is important for generalizing the learned node embeddings that the algorithm has optimized on to a newly observed hypergraph comprising previously unseen nodes, thus, making inductive learning a far more complex problem compared to the transductive learning methods.

In this paper, we address the above mentioned limitations of the existing hypergraph learning methods. We propose a simple yet effective inductive learning framework for hypergraphs that is readily applicable to graphs as well. Our approach relies on neural message passing techniques due to which it can be used on hypergraphs of any degree of cardinality without the need for reduction to graphs. The points below highlight the contributions of this paper:

- We address the challenging problem of representation learning on hypergraphs by proposing HyperSAGE, comprising a message passing scheme which is capable of jointly capturing the intra-relations (within a hyperedge) as well as inter-relations (across hyperedges).

- The proposed hypergraph learning framework is inductive, i.e. it can perform predictions on previously unseen nodes, and can thus be used to model evolving hypergraphs.

- HyperSAGE facilitates neighborhood sampling and provides the flexibility in choosing different ways to aggregate information from the neighborhood.

- HyperSAGE is more stable than state-of-the-art methods, thus provides more accurate results on node classification tasks on hypergraphs with reduced variance in the output.

## 2 RELATED WORK

Learning node representations using graph neural networks has been a popular research topic in the field of geometric deep learning (Bronstein et al., 2017). Graph neural networks can be broadly classified into spatial (message passing) and spectral networks. We focus on a family of spatial message passing graph neural networks that take a graph with some labeled nodes as input and learn embeddings for each node by aggregating information from its neighbors (Xu et al., 2019). Message passing operations in a graph simply propagate information along the edge connecting two nodes. Many variants of such message passing neural networks have been proposed, with some popular ones including Gori et al. (2005); Li et al. (2015); Kipf & Welling (2016); Gilmer et al. (2017); Hamilton et al. (2017).

Zhou et al. (2007) introduced learning on hypergraphs to model high-order relations for semi-supervised classification and clustering of nodes. Emulating a graph-based message passing framework for hypergraphs is not straightforward since a hyperedge involves more than two nodes which makes the interactions inside each hyperedge more complex. Representing a hypergraph with a matrix makes it rigid in describing the structures of higher order relations (Li et al., 2013). On the other hand, formulating message passing on a higher dimensional representation of hypergraph using tensors makes it computationally expensive and restricts it to only small datasets (Zhang et al., 2019). Several tensor based methods do perform learning on hypergraphs (Shashua et al., 2006; Arya et al., 2019), however they are limited to uniform hypergraphs only.

To resolve the above issues, Feng et al. (2019) and Bai et al. (2020) reduce a hypergraph to graph using clique expansion and perform graph convolutions on them. These approaches cannot utilize complete structural information in the hypergraph and lead to unreliable learning performance for e.g. classification, clustering and active learning (Li & Milenkovic, 2017; Chien et al., 2019). Another approach by Yadati et al. (2019), named HyperGCN, replaces a hyperedge with pair-wise weighted edges between vertices (called mediators). With the use of mediators, HyperGCN can be interpreted as an improved approach of clique expansion, and to the best of our knowledge, is also the state-of-the-art method for hypergraph representation learning. However, for many cases such as Fano plane where each hyperedge contains at most three nodes, HyperGCN becomes equivalent to the clique expansion (Dong et al., 2020). In spectral theory of hypergraphs, methods have been proposed that fully exploit the hypergraph structure using non-linear Laplacian operators (Chan et al., 2018; Hein et al., 2013). In this work, we focus on message passing frameworks. Drawing inspiration from GraphSAGE (Hamilton et al., 2017), we propose to eliminate matrix (or tensor) based formulations in our neural message passing frameworks, which not only facilitates utilization of all the available information in a hypergraph, but also makes the entire framework inductive in nature.

## 3 PROPOSED MODEL: HYPERSAGE

The core concept behind our approach is to aggregate feature information from the neighborhood of a node spanning across multiple hyperedges, where the edges can have varying cardinality. Below, we first define some preliminary terms, and then describe our generic aggregation framework. This framework performs message passing at two-levels for a hypergraph. Further, for any graph-structured data, our framework emulates the one-level aggregation similar to GraphSAGE (Hamilton et al., 2017). Our approach inherently allows inductive learning, which makes it also applicable on hypergraphs with unseen nodes.

### 3.1 PRELIMINARIES

**Definition 1** (Hypergraph). *A general hypergraph $\mathcal{H}$ can be represented as $\mathcal{H} = (\mathcal{V}, \mathcal{E}, \mathbf{X})$, where $\mathcal{V} = \{v_1, v_2, ..., v_N\}$ denotes a set of $N$ nodes (vertices) and $\mathcal{E} = \{\mathbf{e}_1, \mathbf{e}_2, ..., \mathbf{e}_K\}$ denotes a set of hyperedges, with each hyperedge comprising a non-empty subset from $\mathcal{V}$. $\mathbf{X} \in \mathbb{R}^{N \times d}$ denote the feature matrix, such that $\mathbf{x}_i \in \mathbf{X}$ is the feature vector characterizing node $v_i \in \mathcal{V}$. The maximum cardinality of the hyperedges in $\mathcal{H}$ is denoted as $M = \max_{\mathbf{e} \in \mathcal{E}} |\mathbf{e}|$.*

Unlike in a graph, the hyperedges of $\mathcal{H}$ can contain different number of nodes and $M$ denotes the largest number. From the definition above, we see that graphs are a special case of hypergraphs with

$M$=2. Thus, compared to graphs, hypergraphs are designed to model higher-order relations between nodes. Further, we define three types of neighborhoods in a hypergraph:

**Definition 2** (Intra-edge neighborhood). *The intra-edge neighborhood of a node $v_i \in \mathcal{V}$ for any hyperedge $\mathbf{e} \in \mathcal{E}$ is defined as the set of nodes $v_j$ belonging to $\mathbf{e}$ and is denoted by $\mathcal{N}(v_i, \mathbf{e})$*

Further, let $E(v_i) = \{\mathbf{e} \in \mathcal{E} \mid v_i \in \mathbf{e}\}$ be the sets of hyperedges that contain node $v_i$.

**Definition 3** (Inter-edge neighborhood). *The inter-edge neighborhood of a node $v_i \in \mathcal{V}$ also referred as its global neighborhood, is defined as the neighborhood of $v_i$ spanning across the set of hyperedges $E(v_i)$ and is represented by $\mathcal{N}(v_i) = \bigcup_{\mathbf{e} \in E(v_i)} \mathcal{N}(v_i, \mathbf{e})$.*

**Definition 4** (Condensed neighborhood). *The condensed neighborhood of any node $v_i \in \mathcal{V}$ is a sampled set of $\alpha \leq |\mathbf{e}|$ nodes from a hyperedge $\mathbf{e} \in E(v_i)$ denoted by $N(v_i, \mathbf{e}; \alpha) \subset \mathcal{N}(v_i, \mathbf{e})$.*

## 3.2 GENERALIZED MESSAGE PASSING FRAMEWORK

We propose to interpret the propagation of information in a given hypergraph as a two-level aggregation problem, where the neighborhood of any node is divided into *intra-edge* neighbors and *inter-edge* neighbors. For message aggregation, we define aggregation function $\mathcal{F}(\cdot)$ as a permutation invariant set function on a hypergraph $\mathcal{H} = (\mathcal{V}, \mathcal{E}, \mathbf{X})$ that takes as input a countable unordered message set and outputs a reduced or aggregated message. Further, for two-level aggregation, let $\mathcal{F}_1(\cdot)$ and $\mathcal{F}_2(\cdot)$ denote the intra-edge and inter-edge aggregation functions, respectively. Schematic representation of the two aggregation functions is provided in Fig.2. Similar to $\mathbf{X}$ we also define $\mathbf{Z}$ as the encoded feature matrix built using the outputs $\mathbf{z}_i$ of aggregation functions. Message passing at node $v_i$ for aggregation of information at the $l^{\text{th}}$ layer can then be stated as

$$\mathbf{x}_{i,l}^{(\mathbf{e})} \leftarrow \mathcal{F}_1\left(\{\mathbf{x}_{j,l-1} \mid v_j \in \mathcal{N}(v_i, \mathbf{e}; \alpha)\}\right), \tag{1}$$

$$\mathbf{x}_{i,l} \leftarrow \mathbf{x}_{i,l-1} + \mathcal{F}_2\left(\{\mathbf{x}_{i,l}^{(\mathbf{e})} \mid v_i \in E(v_i)\}\right), \tag{2}$$

where, $\mathbf{x}_{i,l}^{(\mathbf{e})}$ refers to the aggregated feature set at $v_i$ obtained with intra-edge aggregation for edge $\mathbf{e}$.

The combined two-level message passing is achieved using nested aggregation function $\mathcal{F} = \mathcal{F}_2$. To ensure that the expressive power of a hypergraph is preserved or at least the loss is minimized, the choice of aggregation function should comply with certain properties.

Firstly, the aggregation function should be able to capture the features of neighborhood vertices in a manner that is invariant to the permutation of the nodes and hyperedges. Many graph representation learning methods use permutation invariant aggregation functions, such as $mean$, $sum$ and $max$ functions (Xu et al., 2019). These aggregations have proven to be successful for node classification problems. For the existing hypergraph frameworks, reduction to simple graphs along with a matrix-based message passing framework limits the possibilities of using different types of feature aggregation functions, and hence curtails the potential to explore unique node representations.

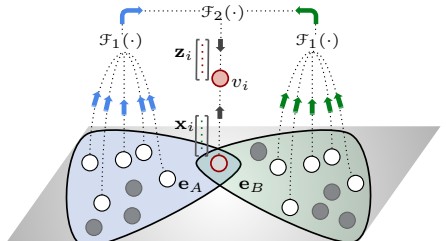

Figure 2: Schematic representation of the two-level message passing scheme of HyperSAGE, with aggregation functions $\mathcal{F}_1(\cdot)$ and $\mathcal{F}_2(\cdot)$. It shows information aggregation from two hyperedges $\mathbf{e}_A$ and $\mathbf{e}_B$, where the intra-edge aggregation is from sampled sets of 5 nodes ($\alpha = 5$) for each hyperedge. For node $v_i$, $\mathbf{x}_i$ and $\mathbf{z}_i$ denote the input and encoded feature vector, respectively.

Secondly, the aggregation function should also preserve the global neighborhood invariance at the 'dominant nodes' of the graph. Here, dominant nodes refer to nodes that contain important features, thereby, impacting the learning process relatively more than their neighbors. The aggregation function should ideally be insensitive to the input, whether the provided hypergraph contains a few large

hyperedges, or a larger number of smaller ones obtained from splitting them. Generally, a hyperedge would be split in a manner that the dominant nodes are shared across the resulting hyperedges. In such cases, global neighborhood invariance would imply that the aggregated output at these nodes before and after the splitting of any associated hyperedge stays the same. Otherwise, the learned representation of a node will change significantly with each hyperedge split.

Based on these considerations, we define the following properties for a generic message aggregation function that should hold for accurate propagation of information through the hypergraphs.

**Property 1** (Hypergraph Isomorphic Equivariance). *A message aggregation function $\mathcal{F}(\cdot)$ is equivariant to hypergraph isomorphism, if for two isomorphic hypergraphs $\mathcal{H} = (\mathcal{V}, \mathcal{E}, \mathbf{X})$ and $\mathcal{H}^* = (\mathcal{V}^*, \mathcal{E}^*, \mathbf{X}^*)$, given that $\mathcal{H}^* = \sigma \bullet \mathcal{H}$, and $\mathbf{Z}$ and $\mathbf{Z}^*$ represent the encoded feature matrices obtained using $\mathcal{F}(\cdot)$ on $\mathcal{H}$ and $\mathcal{H}^*$, the condition $\mathbf{Z}^* = \sigma \bullet \mathbf{Z}$ holds. Here, $\sigma$ denotes a permutation operator on hypergraphs.*

**Property 2** (Global Neighborhood Invariance). *A message aggregation scheme $\mathcal{F}(\cdot)$ satisfies global neighborhood invariance at any node $v_i \in \mathcal{V}$ for a given hypergraph $\mathcal{H} = (\mathcal{V}, \mathcal{E}, \mathbf{X})$ if for any operation $\Gamma(\cdot)$, such that $\mathcal{H}^* = \Gamma(\mathcal{H})$, and $\mathbf{z}_i$ and $\mathbf{z}_i^*$ denote the encoded feature vectors obtained using $\mathcal{F}(\cdot)$ at node $v_i$ on $\mathcal{H}$ and $\mathcal{H}^*$, the condition $\mathbf{z}_i^* = \mathbf{z}_i$ holds. Here $\Gamma(\mathcal{H})$ could refer to operations such as hyperedge contraction or expansion.*

The flexibility of our message passing framework allows us to go beyond the simple aggregation functions on hypergraphs without violating Property 1. We introduce a series of power mean functions as aggregators, which have recently been shown to generalize well on graphs (Li et al., 2020). We perform message aggregation in hypergraphs using these generalized means, denoted by $M_p$ and provide in section 4.2, a study on their performances. We also show that with appropriate combinations of the intra-edge and inter-edge aggregations Property 2 is also satisfied. This property ensures that the representation of a node after message passing is invariant to the cardinality of the hyperedge, i.e., the aggregation scheme should not be sensitive to hyperedge contraction or expansion, as long as the global neighborhood of a node remains the same in the hypergraph.

**Aggregation Functions.** One major advantage of our strategy is that the message passing module is decoupled from the choice of the aggregation itself. This allows our approach to be used with a broad set of aggregation functions. We discuss below a few such possible choices.

*Generalized means.* Also referred to as power means, this class of functions are very commonly used for getting an aggregated measure over a given set of samples. Mathematically, generalized means can be expressed as $M_p = \left(\frac{1}{n} \sum_{i=1}^{n} x_i^p\right)^{\frac{1}{p}}$, where $n$ refers to the number of samples in the aggregation, and $p$ denotes its power. The choice of $p$ allows providing different interpretations to the aggregation function. For example, $p = 1$ denotes arithmetic mean aggregation, $p = 2$ refers to mean squared estimate and a large value of $p$ corresponds to max pooling from the group. Similarly, $M_p$ can be used for geometric and harmonic means with $p \to 0$ and $p = -1$, respectively.

Similar to the recent work of Li et al. (2020), we use generalized means for intra-edge as well as inter-edge aggregation. The two functions $\mathcal{F}_1(\cdot)$ and $\mathcal{F}_2(\cdot)$ for aggregation at node $v_i$ is defined as

$$\mathcal{F}_1^{(i)}(\mathbf{s}) = \left(\frac{1}{|\mathcal{N}(v_i, \mathbf{e})||\mathcal{N}(v_i)|} \sum_{v_j \in \mathcal{N}(v_i, \mathbf{e})} \left(\sum_{m=1}^{|E(v_i)|} \frac{1}{|\mathcal{N}(v_i, \mathbf{e}_m)|}\right)^{-1} \mathbf{x}_j^p\right)^{\frac{1}{p}} \tag{3}$$

$$\mathcal{F}_2^{(i)}(\mathbf{s}) = \left(\frac{1}{|E(v_i)|} \sum_{\mathbf{e} \in E(v_i)} (\mathcal{F}_1(\mathbf{s}))^p\right)^{\frac{1}{p}} \tag{4}$$

where we use 's' for concise representation of the unordered set of input as shown in Eq.1. Here and henceforth in this paper, we remove the superscript index '$(i)$' for the sake of clarity and further occurrences of the two aggregation functions shall be interpreted in terms of node $v_i$. Note that in Eq. 3 and Eq. 4, we have chosen the power term $p$ to be same for $\mathcal{F}_1$ and $\mathcal{F}_2$ so as to satisfy the global neighborhood invariance as stated in Property 2. Note, the scaling term added to $\mathcal{F}_1$ is added to balance the bias in the weighting introduced in intra-edge aggregation due to varying

cardinality across the hyperedges. These restrictions ensure that the joint aggregation $\mathcal{F}_2(\cdot)$ satisfies the property of global neighborhood invariance at all times. Proof of the two aggregations satisfying Property 2 is stated in Appendix B.

**Sampling-based Aggregation.** Our neural message passing scheme provides the flexibility to adapt the message aggregation module to fit the desired computational budget through aggregating information from only a subset $N(v_i, \mathbf{e}; \alpha)$ of the full neighborhood $N(v_i, \mathbf{e})$, if needed. We propose to apply sub-sampling only on the nodes from the training set, and use information from the full neighborhood for the test set. The advantages of this are twofold. First, reduced number of samples per aggregation at training time reduces the relative computational burden. Second, similar to dropout (Srivastava et al., 2014), it serves to add regularization to the optimization process. Using the full neighborhood on test data avoids randomness in the test predictions, and generates consistent output.

### 3.3 INDUCTIVE LEARNING ON HYPERGRAPHS

HyperSAGE is a general framework for learning node representations on hypergraphs, on even unseen nodes. Our approach uses a neural network comprising $L$ layers, and feature-aggregation is performed at each of these layers, as well as across the hyperedges.

Algorithm 1 describes the forward propagation mechanism which implements the aggregation function $\mathcal{F}(\cdot) = \mathcal{F}_2(\cdot)$ described above. At each iteration, nodes first aggregate information from their neighbors within a specific hyperedge. This is repeated over all the hyperedges across all the $L$ layers of the network. The trainable weight matrices $\mathbf{W}^l$ with $l \in L$ are used to aggregate information across the feature dimension and propagate it through the various layers of the hypergraph.

**Generalizability of HyperSAGE.** Hyper-SAGE can be interpreted as a generalized formulation that unifies various existing graph-based as well as hypergraph formulations. Our approach unifies them, identifying each of these as special variants/cases of our method. We discuss here briefly the two popular algorithms.

*Graph Convolution Networks (GCN).* The GCN approach proposed by Kipf & Welling (2016) is a graph-based method that can be derived as a special case of HyperSAGE with maximum cardinality $|M| = 2$, and setting the agggregation function $\mathcal{F}_2 = M_p$ with $p = 1$. This being a graph-based method, $\mathcal{F}_1$ will not be used.

---

**Algorithm 1** HyperSAGE Message Passing

**Input** : $\mathcal{H} = (\mathcal{V}, \mathcal{E}, \mathbf{X})$; depth $L$; weight matrices $\mathbf{W}^l$ for $l = 1 \ldots L$; non-linearity $\boldsymbol{\sigma}$; intra-edge aggregation function $\mathcal{F}_1(\cdot)$; inter-edge aggregation function $\mathcal{F}_2(\cdot)$

**Output:** Node embeddings $\mathbf{z}_i \mid v_i \in \mathcal{V}$

$\mathbf{h}_i^0 \leftarrow \mathbf{x}_i \in \mathbf{X} \mid v_i \in \mathcal{V}$

**for** $l = 1 \ldots L$ **do**
  **for** $\mathbf{e} \in \mathcal{E}$ **do**
    $\mathbf{h}_i^l \leftarrow \mathbf{h}_i^{l-1}$
    **for** $v_i \in \mathbf{e}$ **do**
      $\mathbf{h}_i^l \leftarrow \mathbf{h}_i^l + \mathcal{F}_2^{(i)}(\mathbf{s})$
    **end**
  **end**
  $\mathbf{h}_i^l \leftarrow \sigma(\mathbf{W}^l(\mathbf{h}_i^l / ||\mathbf{h}_i^l||_2)) \mid v_i \in \mathcal{V}$
**end**

$\mathbf{z}_i \leftarrow \mathbf{h}_i^L \mid v_i \in \mathcal{V}$

---

*GraphSAGE.* Our approach, when reduced for graphs using $|M| = 2$, is similar to GraphSAGE. For exact match, the aggregation function $\mathcal{F}_2$ should be one of $mean$, $max$ or $LSTM$. Further, the sampling term $\alpha$ can be adjusted to match the number of samples per aggregation as in GraphSAGE.

## 4 EXPERIMENTS

### 4.1 EXPERIMENTAL SETUP

For the experiments in this paper, we use co-citation and co-authorship network datasets: CiteSeer, PubMed, Cora (Sen et al., 2008) and DBLP (Rossi & Ahmed, 2015). The task for each dataset is to predict the topic to which a document belongs (multi-class classification). For these datasets, $\mathbf{x}_i$ corresponds to a bag of words such that $x_{i,j} \in \mathbf{x}_i$ represents the normalized frequency of occurence of the $j^{th}$ word. Additional details related to the hypergraph topology are presented in Appendix

Table 1: Performance of HyperSAGE and other hypergraph learning methods on co-authorship and co-citation datasets.

| Method | Co-authorship Data | | Co-citation Data | | |
|---|---|---|---|---|---|
| | DBLP | Cora | Pubmed | Citeseer | Cora |
| MLP + HLR | $63.6 \pm 4.7$ | $59.8 \pm 4.7$ | $64.7 \pm 3.1$ | $56.1 \pm 2.6$ | $61.0 \pm 4.1$ |
| HGNN | $69.2 \pm 5.1$ | $63.2 \pm 3.1$ | $66.8 \pm 3.7$ | $56.7 \pm 3.8$ | $\mathbf{70.0 \pm 2.9}$ |
| FastHyperGCN | $68.1 \pm 9.6$ | $61.1 \pm 8.2$ | $65.7 \pm 11.1$ | $56.2 \pm 8.1$ | $61.3 \pm 10.3$ |
| HyperGCN | $70.9 \pm 8.3$ | $63.9 \pm 7.3$ | $68.3 \pm 9.5$ | $57.3 \pm 7.3$ | $62.5 \pm 9.7$ |
| HyperSAGE ($p = 2$) | $71.5 \pm 4.4$ | $69.8 \pm 2.6$ | $71.3 \pm 2.4$ | $59.8 \pm 3.3$ | $62.9 \pm 2.1$ |
| HyperSAGE ($p = 1$) | $77.2 \pm 4.3$ | $\mathbf{72.4 \pm 1.6}$ | $72.6 \pm 2.1$ | $\mathbf{61.8 \pm 2.3}$ | $69.3 \pm 2.7$ |
| HyperSAGE ($p = 0.01$) | $\mathbf{77.4 \pm 3.8}$ | $72.1 \pm 1.8$ | $\mathbf{72.9 \pm 1.3}$ | $61.3 \pm 2.4$ | $68.2 \pm 2.4$ |
| HyperSAGE ($p = -1$) | $70.9 \pm 2.3$ | $67.4 \pm 2.1$ | $68.3 \pm 3.1$ | $59.8 \pm 2.0$ | $62.3 \pm 5.7$ |

A.2. Further, for all experiments, we use a neural network with 2 layers. All models are implemented in Pytorch and trained using Adam optimizer. See Appendix A.2 for implementation details.

## 4.2 SEMI-SUPERVISED NODE CLASSIFICATION ON HYPERGRAPHS

**Performance comparison with existing methods.** We implemented HyperSAGE for the task of semi-supervised classification of nodes on a hypergraph, and the results are compared with state-of-the art methods. These include (a) Multi-layer perceptron with explicit hypergraph Laplacian regularisation (MLP + HLR), (b) Hypergraph Neural Networks (HGNN) (Feng et al., 2019) which uses a clique expansion, and (c) HyperGCN and its variants (Yadati et al., 2019) that collapse the hyperedges using mediators. For HyperSAGE method, we use 4 variants of generalized means $M_p$ with $p = 1, 2, -1$ and $0.01$ with complete neighborhood i.e., $\alpha = |\mathbf{e}|$. For all the cases, 10 data splits over 8 random weight initializations are used, totalling 80 experiments per method and for every dataset. The data splits are the same as in HyperGCN described in Appendix A.1.

Table 1 shows the results obtained for the node classification task. We see that the different variants of HyperSAGE consistently show better scores across our benchmark datasets, except Cora co-citation where no improvement is observed compared to HGNN. Cora co-citation data is relatively small in size with a cardinality of $3.0 \pm 1.1$, and we speculate that there does not exist enough scope of improving with HyperSAGE beyond what HGNN can express with the clique expansion.

For the larger datasets such as DBLP and Pubmed, we see that the improvements obtained in performance with HyperSAGE over the best baselines are 6.3% and 4.3% respectively. Apart from its superior performance, HyperSAGE is also stable, and is less sensitive to the choice of data split and initialization of the weights. This is evident from the scores of standard deviation (SD) for the various experiments in Table 1. We see that the SD scores for our method are lower than other methods, and there is a significant gain in performance compared to HyperGCN. Another observation is that the HyperGCN method is very sensitive to the data splits as well as initializations with very large errors in the predictions. This is even higher for the FastHyperGCN variant. Also, we have found that all the 4 choices of $p$ work well with HyperSAGE for these datasets. We further perform a more comprehensive study analyzing the effect of $p$ on model performance later in this section.

**Stability analysis.** We further study the stability of our method in terms of the variance observed in performance for different ratios of train and test splits, and compare results with that of HyperGCN implemented under similar settings. Fig. 3 shows results for the two learning methods on 5 different train-test ratios. We see that the performance of both models improves when a higher fraction of data is used for training, and the performances are approx-

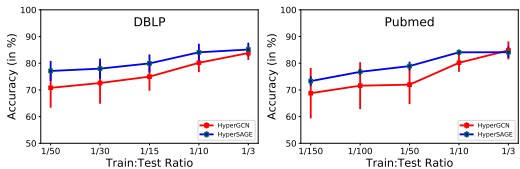

Figure 3: Accuracy scores for HyperSAGE and HyperGCN obtained for different train-test ratios for multi-class classification datasets.

Table 2: Performance of HyperSAGE for multiple values of $p$ in generalized means aggregator ($M_p$) on varying number of neighborhood samples ($\alpha$).

| | DBLP | | | | Pubmed | | | |
|---|---|---|---|---|---|---|---|---|
| | $\alpha = 2$ | $\alpha = 3$ | $\alpha = 5$ | $\alpha = 10$ | $\alpha = 2$ | $\alpha = 3$ | $\alpha = 5$ | $\alpha = 10$ |
| $p = -1$ | 59.6 | 61.2 | 69.9 | 70.9 | 60.1 | 60.2 | 67.9 | 66.4 |
| $p = 0.01$ | 61.2 | 64.8 | 73.1 | **77.4** | 65.5 | 67.4 | **73.4** | 72.9 |
| $p = 1$ | 62.3 | 64.5 | 73.1 | 77.2 | 64.8 | 64.3 | 72.2 | 72.6 |
| $p = 2$ | 63.1 | 63.8 | 71.9 | 71.5 | 63.7 | 63.9 | 70.8 | 71.3 |
| $p = 3$ | 62.7 | 63.6 | 71.3 | 71.4 | 62.2 | 61.3 | 70.1 | 67.9 |
| $p = 5$ | 62.8 | 63.3 | 69.4 | 70.6 | 62.1 | 60.4 | 69.3 | 68.0 |

Table 3: Performance of HyperSAGE and its variants on nodes which were part of the training hypergraph (seen) and nodes which were not part of the training hypergraph (unseen).

| | DBLP | | Pubmed | | Citeseer | | Cora (citation) | |
|---|---|---|---|---|---|---|---|---|
| **Method** | Seen | Unseen | Seen | Unseen | Seen | Unseen | Seen | Unseen |
| MLP + HLR | 64.5 | 58.7 | 66.8 | 62.4 | 60.1 | 58.2 | 65.7 | 64.2 |
| HyperSAGE ($p = 0.01$) | 78.1 | 73.1 | 81.0 | 80.4 | 69.2 | 67.1 | 68.2 | 65.7 |
| HyperSAGE ($p = 1$) | 78.1 | 73.2 | 78.5 | 76.4 | 69.3 | 67.9 | 71.3 | 66.8 |
| HyperSAGE ($p = 2$) | 76.1 | 70.2 | 71.2 | 69.8 | 65.9 | 63.8 | 65.9 | 64.5 |

imately the same at the train-test ratio of 1/3.

However, for smaller ratios, we see that HyperSAGE outperforms HyperGCN by a significant margin across all datasets. Further, the standard deviation for the predictions of HyperSAGE are significantly lower than that of HyperGCN. Clearly, this implies that HyperSAGE is able to better exploit the information contained in the hypergraph compared to HyperGCN, and can thus produce more accurate and stable predictions. Results on Cora and Citeseer can be found in Appendix C.

**Effect of generalized mean aggregations and neighborhood sampling.** We study here the effect of different choices of the aggregation functions $\mathcal{F}_1(\cdot)$ and $\mathcal{F}_2(\cdot)$ on the performance of the model. Further, we also analyze how the number of samples chosen for aggregation affect its performance. Aggregation functions from $M_p$ are chosen with $p = 1, 2, 3, 4, 5, 0.01$ and $-1$, and to comply with global neighborhood invariance, we use aggregation function as in Eq. 4. The number of neighbors $\alpha$ for intra-edge aggregation are chosen to be 2, 3, 5 and 10. Table 2 shows the accuracy scores obtained for different choices of $p$ and $\alpha$ on DBLP and Pubmed datasets. For most cases, higher value of $p$ reduces the performance of the model. For $\alpha = 2$ on DBLP, performance seems to be independent of the choice of $p$. A possible explanation could be that the number of neighbors is very small, and change in $p$ does not affect the propagation of information significantly. An exception is $p = -1$, where the performance drops for all cases. For Pubmed, the choice of $p$ seems to be very important, and we find that $p = 0.01$ seems to fit best.

We also see that the number of samples per aggregation can significantly affect the performance of the model. For DBLP, model performance increases with increasing value of $\alpha$. However, for Pubmed, we observe that performance improves up to $\alpha = 5$, but then a slight drop is observed for larger sets of neighbors. Note that for Pubmed, the majority of the hyperedges have cardinality less than or equal to 10. This means that during aggregation, information will most often be aggregated from all the neighbors, thereby involving almost no stochastic sampling. Stochastic sampling of nodes could serve as a regularization mechanism and reduce the impact of noisy hyperedges. However, at $\alpha = 10$, it is almost absent, due to which the noise in the data affects the performance of the model which is not the case in DBLP.

### 4.3 INDUCTIVE LEARNING ON EVOLVING GRAPHS

For inductive learning experiment, we consider the case of evolving hypergraphs. We create 4 inductive learning datasets from DBLP, Pubmed, Citeseer and Core (co-citation) by splitting each

of the datasets into a train-test ratio of 1:4. Further, the test data is split into two halves: *seen* and *unseen*. The seen test set comprises nodes that are part of the hypergraph used for representation learning. Further, unseen nodes refer to those that are never a part of the hypergraph during training. To study how well HyperSAGE generalizes for inductive learning, we classify the unseen nodes and compare the performance with the scores obtained on the seen nodes. Further, we also compare our results on unseen nodes with those of MLP+HLR. The results are shown in Table 3. We see that results obtained with HyperSAGE on unseen nodes are significantly better than the baseline method. Further, these results seem to not differ drastically from those obtained on the seen nodes, thereby confirming that HyperSAGE can work with evolving graphs as well.

## 5 CONCLUSION

We have proposed HyperSAGE, a generic neural message passing framework for inductive learning on hypergraphs. The proposed approach fully utilizes the inherent higher-order relations in a hypergraph structure without reducing it to a regular graph. Through experiments on several representative datasets, we have shown that HyperSAGE outperforms the other methods for hypergraph learning. Several variants of graph-based learning algorithm such as GCN and GraphSAGE can be derived from the flexible aggregation and neighborhood sampling framework, thus making HyperSAGE a universal framework for learning node representations on hypergraphs as well as graphs.

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

APPENDICES

## A EXPERIMENTS: ADDITIONAL DETAILS

We perform multi-class classification on co-authorship and co-citation datasets, where the task is to predict the topic (class) for each document.

### A.1 DATASET DESCRIPTION

Hypergraphs are created on these datasets by assigning each document as a node and each hyperedge represents (a) all documents co-authored by an author in co-authorship dataset and (b) all documents cited together by a document in co-citation dataset. Each document (node) is represented by bag-of-words features. The details about nodes, hyperedges and features is shown in Table 4. We use the same dataset and train-test splits as provided by Yadati et al. (2019) in their publically available implementation [1].

Table 4: Details of real-world hypergraph datasets used in our work

|  | Co-authorship Data | | Co-citation Data | | |
| --- | --- | --- | --- | --- | --- |
|  | DBLP | Cora | Pubmed | Citeseer | Cora |
| Nodes ($|\mathcal{V}|$) | 43413 | 2708 | 19717 | 3312 | 2708 |
| Hyperedges ($|\mathcal{E}|$) | 22535 | 1072 | 7963 | 1079 | 1579 |
| average hyperedge size | 4.7$\pm$6.1 | 4.2$\pm$4.1 | 4.3 $\pm$ 5.7 | 3.2$\pm$2.0 | 3.0 $\pm$ 1.1 |
| number of features, $|\mathbf{x}|$ | 1425 | 1433 | 500 | 3703 | 1433 |
| number of classes | 6 | 7 | 3 | 6 | 7 |

### A.2 IMPLEMENTATION DETAILS

We use the following set of hyperparameters similar to the prior work by Kipf & Welling (2016) for all the models.

- hidden layer size: 32
- dropout rate: 0.5
- learning rate: 0.01
- weight decay: 0.0005
- number of training epochs: 150
- $\lambda$ for explicit Laplacian regularisation: 0.001

## B CHOICE OF INTER-EDGE AND INTRA-EDGE AGGREGATIONS

*Proof.* For any given hypergraph $\mathcal{H}_1 = (\mathcal{V}, \mathcal{E}_1, \mathbf{X})$, let $v_i$ denote a node at which global neighborhood equivariance exists. The aggregation output $\mathcal{F}_1(\mathbf{s})$ at $v_i$ can then be written using generalized means $M_p$ as

$$\mathcal{F}_1(\mathbf{s}) = \left( \frac{1}{|\mathcal{N}(v_i, \mathbf{e})|} \sum_{v_j \in \mathcal{N}(v_i, \mathbf{e})} \mathbf{x}_j^{p_1} \right)^{\frac{1}{p_1}}. \tag{5}$$

To reiterate here, $\mathbf{s}$ denotes the unordered set of input as shown in Eq. 5. Further, the inter-edge aggregation $\mathcal{F}_2(\cdot)$ can be stated as

---

[1]HyperGCN Implementation: https://github.com/malllabiisc/HyperGCN

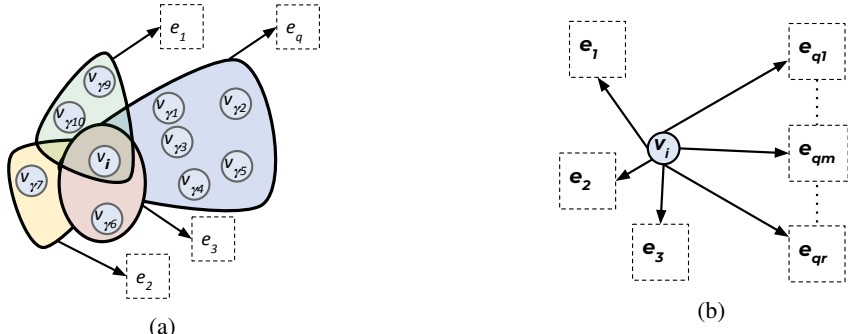

Figure 4: (a) Example showing node $v_i$ shared across 4 hyperedges. (b) Hyperedge $e_q$ is split into $r$ hyperedges to reduce the cardinality of $e_q$. Note that the global neighborhood of $v_i$ still remains the same, however its intra-edge neighborhood has changed due to such splitting.

$$\mathcal{F}_2(\mathbf{s}) = \left( \frac{1}{|E(v_i)|} \sum_{\mathbf{e} \in E(v_i)} \left( \frac{1}{|\mathcal{N}(v_i, \mathbf{e})|} \sum_{v_j \in \mathcal{N}(v_i, \mathbf{e})} \mathbf{x}_j^{p_1} \right)^{\frac{p_2}{p_1}} \right)^{\frac{1}{p_2}} \tag{6}$$

This equation can be rewritten as

$$\mathcal{F}_2(\mathbf{s}) = \left( \frac{1}{|E(v_i)|} \left( \left( \frac{1}{|\mathcal{N}(v_i, \mathbf{e}_q)|} \sum_{v_j \in \mathcal{N}(v_i, \mathbf{e}_q)} \mathbf{x}_j^{p_1} \right)^{\frac{p_2}{p_1}} + \sum_{\mathbf{e} \in E(v_i), \mathbf{e} \neq \mathbf{e}_q} \left( \frac{1}{|\mathcal{N}(v_i, \mathbf{e})|} \sum_{v_j \in \mathcal{N}(v_i, \mathbf{e})} \mathbf{x}_j^{p_1} \right)^{\frac{p_2}{p_1}} \right) \right)^{\frac{1}{p_2}} \tag{7}$$

Further, let

$$\Psi = \sum_{\mathbf{e} \in E(v_i), \mathbf{e} \neq \mathbf{e}_q} \left( \frac{1}{|\mathcal{N}(v_i, \mathbf{e})|} \sum_{v_j \in \mathcal{N}(v_i, \mathbf{e})} \mathbf{x}_j^{p_1} \right)^{\frac{p_2}{p_1}}, \tag{8}$$

then Eq. 7 can be rewritten as

$$\mathcal{F}_2(\mathbf{s}) = \left( \frac{1}{|E(v_i)|} \left( \left( \frac{1}{|\mathcal{N}(v_i, \mathbf{e}_q)|} \sum_{v_j \in \mathcal{N}(v_i, \mathbf{e}_q)} \mathbf{x}_j^{p_1} \right)^{\frac{p_2}{p_1}} + \Psi \right) \right)^{\frac{1}{p_2}} \tag{9}$$

Let us assume now that hyperedge $\mathbf{e}_q$ is split into $r$ hyperedges given by $E(v_i, \mathbf{e}_q) = \{\mathbf{e}_{q_1}, \mathbf{e}_{q_2} \ldots \mathbf{e}_{q_r}\}$. Stating the aggregation on the new set of hyperedges as $\tilde{\mathcal{F}}_2(\mathbf{s})$, we assemble the contribution from this new set of hyperedges with added weight terms $w_j$ as stated below.

$$\tilde{\mathcal{F}}_2(\mathbf{s}) = \left( \frac{1}{|E(v_i)|} \left( \sum_{\mathbf{e} \in E(v_i, \mathbf{e}_q)} \left( \frac{1}{|\mathcal{N}(v_i, \mathbf{e})|} \sum_{v_j \in \mathcal{N}(v_i, \mathbf{e})} w_j \mathbf{x}_j^{p_1} \right)^{\frac{p_2}{p_1}} + \Psi \right) \right)^{\frac{1}{p_2}} \tag{10}$$

For the property of global neighborhood invariance to hold at $v_i$, the following condition should be satisfied: $\mathcal{F}_2(v_i) = \tilde{\mathcal{F}}_2(v_i)$. Based on this, we would like to solve for the weights $w_j$. For this, we equate the two terms and obtain

$$\left( \frac{1}{|\mathcal{N}(v_i, \mathbf{e}_q)|} \sum_{v_j \in \mathcal{N}(v_i, \mathbf{e}_q)} \mathbf{x}_j^{p_1} \right)^{\frac{p_2}{p_1}} = \sum_{\mathbf{e} \in E(v_i, \mathbf{e}_q)} \left( \frac{1}{|\mathcal{N}(v_i, \mathbf{e})|} \sum_{v_j \in \mathcal{N}(v_i, \mathbf{e})} w_j \mathbf{x}_j^{p_1} \right)^{\frac{p_2}{p_1}} \tag{11}$$

We further solve for the variables $p_1$, $p_2$ and $w_j$ where Eq. 11 holds. For the sake of clarity, we first simplify Eq. 11 using the following substitutions: $\alpha = \frac{p_2}{p_1}$, $\beta = \frac{1}{|\mathcal{N}(v_i, \mathbf{e}_q)|}$ and $\beta_{mj} = \frac{w_j}{|\mathcal{N}(v_i, \mathbf{e}_m)|}$, where the index $m$ here is used to refer to the $m^{\text{th}}$ hyperedge from among the $r$ hyperedges obtained on splitting $\mathbf{e}_q$. Further, let $z_j = \mathbf{x}_j^{p_1}$ for $v_j \in \mathcal{N}(v_i, \mathbf{e}_q)$ and $z_{mj} = \mathbf{x}_j^{p_1}$ for $v_j \in \mathcal{N}(v_i, \mathbf{e}_m)$ and $\mathbf{e}_m \in E(v_i, \mathbf{e}_q)$.

Based on these substitutions, Eq. 11 can be restated as

$$
\begin{aligned}
\beta^\alpha (z_1 + z_2 + \ldots + z_N)^\alpha &= (\beta_{11} z_1 + \beta_{12} z_2 + \ldots + \beta_{1j} z_j + \ldots + \beta_{1N} z_N)^\alpha \\
&+ (\beta_{21} z_1 + \beta_{22} z_2 + \ldots + \beta_{2j} z_j + \ldots + \beta_{2N} z_N)^\alpha + \\
&\vdots \\
&+ (\beta_{r1} z_1 + \beta_{r2} z_2 + \ldots + \beta_{rj} z_j + \ldots + \beta_{rN} z_N)^\alpha.
\end{aligned}
\tag{12}
$$

We seek general solutions for $w_j$ and $\alpha$ which holds for all values of $z_j \in [0,1]$ since every element in the normalized feature vectors $\mathbf{x}_j$ lies in $[0,1]$.

For a generalized solution, the coefficients of $z_j$ on the right should be equal to the coefficient of $z_j$ on the left. The term on the left can be reformulated as

$$
\beta^\alpha (z_1 + z_2 + \ldots + z_N)^\alpha = \beta^\alpha (z_1 + (z_2 + z_3 + \ldots + z_N))^\alpha
\tag{13}
$$

Consider the case when $|z_1| \le |z_2 + z_3 + \ldots|$, we expand Eq. 13. using binomial expansion for real co-efficients,

$$
\begin{aligned}
\beta^\alpha (z_1 + (z_2 + z_3 + \ldots))^\alpha &= \beta^\alpha (\binom{\alpha}{0} z_1^\alpha + \binom{\alpha}{1} z_1^{\alpha-1} (z_2 + z_3 + \ldots + z_N) + \\
&\vdots \\
&+ \binom{\alpha}{\alpha - 1} z_1 (z_2 + z_3 + \ldots + z_N)) \\
&= \beta^\alpha (z_1^\alpha + \alpha(z_1^{\alpha-1} z_2 + z_1^{\alpha-1} z_3 + \ldots + z_1^{\alpha-1} z_N) + \\
&\vdots \\
&+ \alpha z_1 (z_2 + z_3 + \ldots + z_N)^{\alpha-1})
\end{aligned}
\tag{14}
$$

Without any loss of generality, we consider splitting of hyperedge $e_q$ into $r$ hyperedges such that nodes $v_{\gamma_1}$ and $v_{\gamma_2}$ are not contained in the same hyperedge anymore. This implies that RHS in Eq. 14 should not contain product terms of $z_1$ and $z_2$. Hence, the term $z_1^{\alpha-1} z_2$ should be such that

$$
\alpha - 1 = 0 \Rightarrow \alpha = 1 \Rightarrow p1 = p2
\tag{15}
$$

Putting $\alpha = 1$ and comparing the coefficients in Eq.12, we get

$$
\beta = \beta_{11} + \beta_{12} + \ldots + \beta_{21} + \beta_{22} \ldots + \beta_{r1} + \beta_{r2} + \ldots
$$

$$
\frac{1}{|\mathcal{N}(v_i, \mathbf{e}_q)|} = \sum_{m=1}^{r} \frac{w_j}{|\mathcal{N}(v_i, \mathbf{e}_m)|}
\tag{16}
$$

$$
w_j = \frac{1}{|\mathcal{N}(v_i, \mathbf{e}_q)|} * \left( \sum_{m=1}^{r} \frac{1}{|\mathcal{N}(v_i, \mathbf{e}_m)|} \right)^{-1}
\tag{17}
$$

Thus, if an edge $\mathbf{e}_q$ is split into multiple edges $E(v_i, \mathbf{e}_q)$, then for the two aggregations to hold, the conditions are $p_1 = p_2$ and $w_j = \frac{1}{|\mathcal{N}(v_i, \mathbf{e}_q)|} * \left( \sum_{m=1}^{r} \frac{1}{|\mathcal{N}(v_i, \mathbf{e}_m)|} \right)^{-1}$ $\forall$ $\mathbf{e} \in E(v_i, \mathbf{e}_q)$.

While we provide above a description related to splitting a certain hyperedge $\mathbf{e}_q$ into $r$ hyperedges, the derived results can be used to compute global neighborhood itself on any given node $v_i$. Similar to $\mathbf{e}_q$ above, node $v_i$ together with its global neighborhood (counted as $\mathcal{N}(v_i)$) can be interpreted as a virtual hyperedge that has been split into a number of hyperedges that actually exist and contain $v_i$. These resultant hyperdges are equivalent to the $r$ hyperdges obtained after splitting, as stated above.

# C    STABILITY TEST ON CORA AND CITESEER

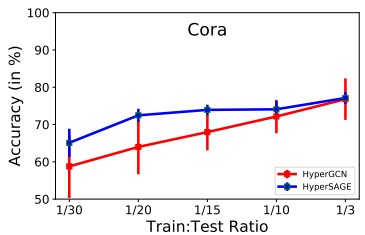 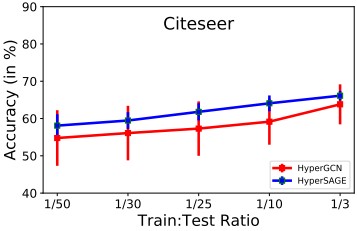

Figure 5: Results on cora and citeseer for multiple train test ratio

