# OpenReview forum: "HyperSAGE: Generalizing Inductive Representation Learning on Hypergraphs"
_ICLR.cc/2021/Conference — Reject_

### Official Review · AnonReviewer3 · 2020-10-25
**An extension of GraphSage towards Hypergraphs**

**Rating:** 6
**Confidence:** 3

**Review:**

Summary
In this paper, the authors study the problem of learning node embeddings for hypergraphs. While most of the existing studies consider reducing hyper-graphs into graphs, this paper studies learning embeddings directly on the hypergraphs using two stages of aggregations / sampling. The efficacy of the proposed method is illustrated in a semi-supervised as well as inductive setting, where the method achieves better performances than the baselines.

Originality.
This paper may be interpreted as extension of GraphSage towards hypergraphs. The main contribution is in the aggregation function, which has two levels: one over intra-edge neighbourhood, and other over inter-edge neighbourhood. This method is thus suitable for both transductive as well as inductive settings.

Significance.
The experimental results show that HyperSage achieves better performance than HyperGCN in the semi-supervised tasks, while also extends to inductive settings and compares favourably against the MLP+HLR baseline.  In this aspect, the results are significant and relevant to the community.

Clarity:
This paper is well written and easy to follow. The relevant baselines and related methods are discussed appropriately. There are quite a few typos and some lack of clear notations, which are listed below:

	- Intra edge neighborhood - definition unclear - How is v_i used in the definition of N(v_i, e).

	- Typo in eqn 2, should have been e \in E(v_i)

	- Typo in eqn 3, the second equation has ratio of two sets, which ideally should have been ratio of cardinality of two sets.

	- Algorithm 1, line 4, initialization of h_i^l from h_i^{l-1} should be outside the for loop, otherwise it may get reset again repeatedly.


Questions to authors:

	- HyperSage has less deviation in the results than other methods such as HyperGCN. Any reasons ?

	- In the stability analysis paragraph, it is shown that as the train to test ratio improves to 1/3, HyperGCN and HyperSage have nearly same performances. Would be interesting to see how it behaves as we increase the training ratio further. Also, any insights into why this behaviour happens would be useful.

	- Also, what is the specific train-test ratio which is used in reporting Table 1? It would be better to include that in the main paper, since there is a discussion on train-test already present.

        - What are the other possible baselines for comparison in the inductive setting (Bai.et.al (Pattern Recognition, 2020) ? )

        - From the results presented in Tables 1,2,3, it appears that the best performance is shared between p = 0.001 (equivalent to that  of Geometric mean aggregator) and p = 1(arithmetic mean aggregator). Any explanations of this would be useful.


Pros:

       - An effective and simple extension of GraphSage towards HyperGraphs, which is suitable for inductive and transductive settings.

       - Experiment results illustrating better performances than the baselines.

Cons:

       - The novelty of the proposed method is limited to that of the aggregation functions.

       - Some of the results mentioned lack explanation.

---

> ### Author Response · Authors · 2020-11-15
> **Response to Reviewer #4 Part1/2**
>
> We would like to thank the reviewer for excellent comments and suggestions that helped us further improve the paper.
>
> *Q- Intra edge neighborhood - definition unclear - How is v_i used in the definition of $\mathcal{N}(v_i, \mathbf{e})$.*
> R- Thank you for pointing to a definition creating lack of clarity. The intra-edge neighborhood of a node v_i are all the nodes which co-occur with $v_i$ in a specific hyperedge $\mathbf{e}$. For example, given hyperedge $\mathbf{e}= \[v_1,v_2,v_3\]$, then $\mathcal{N}(v_1, \mathbf{e})= [v_2,v_3]$. We will add this small example to depict the Intra edge neighborhood more clearly. Let us know if it needs to be clarified further.
>
> *Q- HyperSage has less deviation in the results than other methods such as HyperGCN. Any reasons ?*
> R- Figure 3 and Figure 4 (in Appendix) summarizes an experiment with different train-test ratios, conducted to show that (a) HyperSAGE performs better than HyperGCN even with less training data, (b) the variance in accuracy of HyperSAGE is much lower as compared to HyperGCN, showing a more stable result. HyperSAGE yields more stable results than HyperGCN chiefly because the set of neighbors used for message passing at each epoch is constant (either all the neighborhoods or a fixed number of samples) in case of HyperSAGE. However, in HyperGCN, for each epoch, the furthest away (i.e. maximally distant) neighboring node is chosen with higher weight for message passing while other nodes are given lower weights. In this way, there is a substantial drop in the uniformity of information that is propagated in each epoch, making the approach particularly sensitive to different initialization parameters of the model. Such crude approximation of neighborhood dents the performance of the model significantly and can only perform at par when provided with substantial amounts of training data.
>
> *Q-In the stability analysis paragraph, it is shown that as the train to test ratio improves to 1/3, HyperGCN and HyperSage have  nearly the same performances. Would be interesting to see how ...*
> R- Thank you! We did perform experiments beyond the train to test ratio of 1:3. We found out that after this ratio the models perform comparably. Below are some results obtained for train-test ratio of 1:2 by different models:
>
> |        |HyperGCN           |    HyperSAGE |
> | ------------- |:-------------:| -----:|
> | DBLP      | 83.2   | 84.1  |
> | Pubmed      | 86.9      | 87.3   |
> | Cora | 82.5      | 82.4     |
> |Citeseer | 66.8               | 67.4       |
>
> This further supports our point from the previous response  needs substantially more training data for a similar level of performance. This insight is important as a vast majority of related work attempts to reduce hypergraphs to graphs for learning, which inevitably leads to a loss of information. In this paper we demonstrate that applying our simple message passing scheme facilitates utilization of the full expressive power of hypergraphs.
> If the reviewer believes that that it would contribute to improved clarity, we could e.g. modify Figure 3 by including additional train-test ratios.
>
> *Q- Also, what is the specific train-test ratio which is used in reporting Table 1?*
> R- Thank you for pointing this out. Originally we included these details in the supplementary material (cf. Section A.1), but for improved readability we will move it to the main text. We used exactly the same splits as reported in HyperGCN for fair comparison in Table 1.
>
> *Q- What are the other possible baselines for comparison in the inductive setting (Bai.et.al (Pattern Recognition '20) )*
> R- Thank you for bringing this paper to our attention. Although very different from our approach as it uses an attention module, the idea of utilizing hypergraph attention sounds very interesting although not very suitable for inductive setting. Judging by the metadata on Elsevier website, it has been available online since 14 September 2020 and it is scheduled to appear in February 2021 volume of Pattern Recognition journal. Since it was apparently accepted for publication at a peer-reviewed venue and available online just two weeks before ICLR’21 deadline. However, we will carefully study their proposed approach and, if it appears to be applicable to our use case of inductive learning on evolving graphs, by January include the results in e.g. Section 4.3 and Table 3 of the final paper draft.
> We would like to note again that at the moment when we submitted this paper, HyperGCN was the best performing relevant method. Unfortunately, there are very few inductive graph learning approaches, and even less (or practically none) in case of hypergraphs. Another reviewer pointed us to a potentially relevant hypergraph learning approach that was recently published at Sets and Partitions workshop in NeurIPS’19, which we will study and, if deemed indeed relevant, include in Section 4.3 as well.

---

> > ### Author Response · Authors · 2020-11-15
> > **Response to Reviewer #4 Part2/2**
> >
> > *Q- From the results presented in Tables 1,2,3, it appears that the best performance is shared between p = 0.001 (equivalent to that  of Geometric mean aggregator) and p = 1(arithmetic mean aggregator). Any explanations of this would be useful.*
> > R- Thank you for a very interesting observation. We conjecture that the reason why the best performance is shared between geometric mean and arithmetic mean lies in data distribution. In general, the higher order means are designed to assign higher weight to larger values and reduce the impact of the contributions from smaller values. Such higher values were possibly not beneficial here, which is why the geometric and arithmetic means are preferred.
> >
> > As for why geometric mean yields the best performance on two datasets and the arithmetic mean is the best performer on the other two, we would like to make a point on their size and complexity. DBLP and Pubmed are relatively larger and noisy datasets. In addition, the cardinality of the hyperedges in these datasets is high. These factors lead to large contrasts in feature values across the nodes. To capture the effect of such patterns geometric means are a better choice since these are designed to work well with exponentially distributed data. On the other hand, the other two datasets are small and contain hyperedges with very low cardinality. Reducing noise and large contrasts in the distribution, makes arithmetic mean a better choice in this case over geometric mean. We will include a condensed form of the response in the next paper draft.
> >
> > *Q- The novelty of the proposed method is limited to that of the aggregation functions.*
> > R- We would like to point the reviewer to the last paragraph of Introduction where we summarize contributions of this paper. While, indeed, aggregation functions are an important contribution of the proposed approach, it is not the only one.  We conjecture that the simplicity of our framework and its reliance on widely-deployed neural message passing schemes are actually an advantage that will allow for easier adaptation and deployment of existing graph learning approaches to hypergraphs.
> >
> > In addition, our approach is addressing a fundamental problem that was frequently overlooked, misinterpreted or oversimplified in representation learning literature. Since the graph learning approaches are readily available, a vast majority of related work attempts to reduce hypergraphs to graphs for learning, which inevitably leads to a loss of information. In this paper we demonstrate that  applying such a simple message passing scheme facilitates utilization of the full expressive power of hypergraphs.
> >
> > Contrary to most related work on hypergraph learning, here we respect the fact that hypergraph is a generalization of a graph and show that some popular graph-based approaches can be derived from our proposed HyperSAGE framework by selecting an appropriate maximum cardinality (cf. e.g. Section 3.3). We have also provided two important propositions, which a hypergraph based learning model should adhere to.
> >
> > To summarize in slightly different terms, the novelty of our proposed framework lies in (a) its generalizability, as multiple existing algorithms can be derived as a special case of HyperSAGE, (b) the flexibility it provides with two-level aggregation and sampling approach, and (c) the stability in the performance even while using less training data then the state of the art alternatives.
> >
> > *Q- Some of the results mentioned lack explanation.*
> > R- Thank you again for pointing this out. We will extend discussion in Section 4 with the insights from responses to the reviewer’s detailed comments above.

---

### Official Review · AnonReviewer1 · 2020-10-28
**Better and more formal presentation is required; recent literature needs to be propertly cited**

**Rating:** 4
**Confidence:** 4

**Review:**

This paper introduces HyperSAGE, a method to address the problem of representation learning for hypergraphs.

The chosen approach consists in defining aggregation functions adapted to this specific type of data. It allows inductive learning while methods based on reduction to regular graphs are inherently transductive.

The algorithm introduced in this manuscript achieves good accuracy scores compared to other recent methods for hypergraphs on several real datasets. It is also more stable.


Main concerns:

1/ The paper deserves a better presentation: the authors should be more precise and also more formal at some places of the paper, in particular to describe the problem under consideration and the model they develop.

For example:
- The reader is expected to guess that nodes have features that are stored in a matrix (which supposes that all the nodes have the same features and are real-valued). Indeed, hypergraphs are only defined through their topology (see Def. 1). The feature matrix is then introduced without any detail and explanation on what it models (see Subsection 3.2).

- The concept of aggregation function should be introduced in the paper (for example at the beginning of Subsection 3.2). The two aggregation functions are not defined. In eq. (1) and (2), the reader can understand that the input of an aggregation function
is a subset of features. In Prop. 1, the input is a hypergraph (which has no feature if one reads Def. 1 to the letter).  In eq. (3), the input is a node.

- Property 2 should be written in a formal way. In particular, I do not fully understand the relation between what is claimed to be proven in Appendix B and Prop. 2. In addition, if I understand correctly, the proof does not show that p_1 is equal to p_2:
"we first assume p_1=p_2" under eq. (10), while the case p_1\neq p_2 is not dealt with. Finally, an additional condition seems to be assumed in the proof while not mentioned in the main document (see sentence under eq. (11)).

For these reasons, Sections 2 and 3 (as well as Appendix B) are difficult to read and understand.

2/ It seems to me that the recent literature on hypergraph-based learning approaches is not properly cited. The authors should have a look at the following papers, in particular at the second one that proposes inductive (and transductive) learning methods for hypergraphs.
- "Hyperedge2vec: Distributed representations for hyperedges", A Sharma, S Joty, H Kharkwal, J Srivastava (Preprint, 2018)
- "Deep Hyperedges: a Framework for Transductive and Inductive Learning on Hypergraphs", J Payne, published in Sets & Partitions Workshop at NeurIPS 19.


Questions:
- What is a uniform hypergraph as mentioned in Section 2?
- What is the global neighborhood of a node?
Is that the union of intra-edge and inter-edge neighborhoods?


Few typos:
- page 3: space missing
active learning etc.(Li
- page 5 (property 1)
isomporphism
- page 5
The two functions (...) is defined

---

> ### Author Response · Authors · 2020-11-19
> **Response to Reviewer #3, Part 1/2**
>
> Thank you for the constructive review! As this an open discussion phase, we would value your feedback on our responses to better understand how we can further improve the paper. We would like to thank the reviewer for making a good point and we admit that we could better clarify and formalize some statements in the paper. We carefully considered the reviewer's comments below and addressed such possible sources of misunderstanding as detailed in our responses for definition of some basic concepts  and elaborating the proof for a more generic case.
>
> *Q- The reader is expected to guess that nodes have features ....*
> R - We agree that for  standalone representation of HyperSAGE, these notations should be clearly defined. We now clarify this in Definition 1. Further, we have improved the definition of feature matrix in Section 3.1 making it clear that feature matrix $\mathbf{X} \in \mathbb{R}^{N \times d}$ comprises feature vectors $\mathbf{x}_i$. The terms $N$ and $d$ are defined in Definition 1.
>
> Regarding a more general/conceptual remark about the purpose of feature matrix in our framework, we would like to note that the very advantage of using graphs (and hypergraphs as their generalized forms) is the flexibility they provide in modeling complex real-world problems using their two integral components - structure (topology) and content (features on the nodes). Depending on the dataset and problem at hand, some relations are better modelled “directly” with the (hyper)graph topology, while the other relations are easier to capture in the features and their similarities.
>
> For our experiments on citation and co-authorship networks,  hypergraphs were created on these datasets by assigning each document as a node and each hyperedge comprised  (a) all documents co-authored by an author in co-authorship dataset and (b) all documents cited together by a document in co-citation dataset. Each document (node) was represented by bag-of-words features. These details about nodes, hyperedges and features were stated before in Appendix A.1 and Table 4. In the updated draft, we have now explicitly mentioned the choice of feature vectors in Section 4.1.
>
> *Q-  The concept of aggregation function should be introduced... *
> R - Thank you for pointing to a possible inconsistency in terminology. In e.g. Abstract and penultimate sentence of Introduction we mention “different ways of aggregating neighborhood information.” actually referring to the aggregation functions. We have defined the aggregation function now in Section 3.2.  Further, for intuitive understanding we have added reference to Figure 2 of the paper. With these changes, we hope to have sufficiently answered your concern. Please let us know if any more improvements should be made. Further, based on your suggestion we have improved Definition 1 to clarify that a (hyper)graph encodes both structure and content (i.e. includes features on the nodes as well).
>
> *Q- Property 2 should be written in a formal way. *
> R- Thank you. We have improved the text in Property 2, making it more formal and aligning it better with the proof from Appendix B.
>
> *Q- Related Work*
> R- Thank you for pointing us to these two interesting works, we were aware of the work by Sharma et al.,  however since it was not accepted after peer review we refrained from building up our related work on this article. However, after thoroughly reading the reviews of this paper we agree on its inclusion and we will include it. Further, the second which was recently published at Sets and Partitions workshop in NeurIPS’19 article. It again converts a hypergraph to graph by performing random walks which we have argued is not the optimal choice. The implementation of the methodology is not clearly stated by the author yet. We would like to note again that at the moment when we submitted this paper, HyperGCN was the best performing relevant method. Unfortunately, there are very few inductive graph learning approaches, and even less (or practically none) in case of hypergraphs. Based on your recommendations as well as the other reviewer we will be expanding the related work section in the final draft.
>
> *Q- What is a uniform hypergraph?*
> R- Thanks for pointing this out. In a uniform hypergraph all the hyperedges have same cardinality i.e. each hyperedge contains same number of nodes. We will include this line in Section 2.
>
> *Q- What is the global neighborhood of a node? Is that the union of intra-edge and inter-edge neighborhoods?*.
> We would like to point to definition 3 of the paper where we had stated that inter-edge neighborhood is referred to as global neighborhood and that it is the union of intra-edge neighborhood of a node.

---

> > ### Author Response · Authors · 2020-11-21
> > **Response to Reviewer #3, Part 2/2**
> >
> > *Q- In addition, if I understand correctly, the proof does not show that p_1 is equal to p_2: "we first assume $p_1=p_2$" under eq. (10), while the case $p_1\neq p_2$ is not dealt with. Finally, an additional condition seems to be assumed in the proof while not mentioned in the main document (see sentence under eq. (11)).*
> > R- Thank you for this remark. We apologize for ignoring the case where p1 is not equal to p2. We have now provided a more complete proof in the updated draft.  We have alleviated the use of the additional condition and made the proof more generic. We have included a figure for more clarity and now a hyperedge can be split into multiple hyperedges with any amount of vertices in them.  Accordingly, we have updated the draft to reflect this. Please let us know in case there is something till unaddressed in 3b and 3c combined.
> >
> > Finally, we reiterate that HyperSAGE provides a hypergraph message passing framework which is more generic, flexible, uses all the information within a hypergraph and has lesser variance while performing node classification as compared to existing methods. In addition, it adheres to two important properties that we have defined, which is extremely essential to consider while designing any message passing algorithm in hypergraphs.

---

### Official Review · AnonReviewer4 · 2020-10-29
**A simple and flexible message passing method for hypergraphs**

**Rating:** 5
**Confidence:** 3

**Review:**

pros.
-The paper proposed a simple and flexible message passing method for hypergraphs.
-By performing semi-supervised node classification on 4 popular datasets, authors show their method outperforms existing methods.
-The paper is well-organized and clearly written. To the best of my knowledge, the method is technically sound.

cons
-The novelty of the paper seems limited. The proposed method defined in Equ. (1) and (2) is a straightforward extension of the message-passing method for ordinary graphs.
- In Equ. (3): N(v, e) -> |N(v_i, e)|, N(v_i) -> |N(v_i)|

comments
From Tab. (1),  it seems the reported performance of hypergraph-based methods is not better than the results of GAT, GCN. So, why should we use hypergraph-based methods?

---

> ### Author Response · Authors · 2020-11-13
> **Response to Reviewer #2**
>
> We would like to thank the reviewer for valuable comments and suggestions. Below we provide detailed responses to reviewer comments.
>
> *Q:The novelty of the paper seems limited. The proposed method defined in Equ. (1) and (2) is a straightforward extension of the message-passing method for ordinary graphs.*
> R: We agree with the reviewer that one of the primary contributions of this paper is the extension of the well-known message passing scheme from graphs to hypergraphs. We conjecture that the simplicity of our framework and its reliance on widely-deployed neural message passing scheme are actually an advantage that will allow for easier adaptation and deployment of existing graph learning approaches to hypergraphs.
> In addition, our approach is addressing a fundamental problem that was frequently overlooked, misinterpreted or oversimplified in representation learning literature. Since the graph learning approaches are readily available, a vast majority of related work attempts to reduce hypergraphs to graphs for learning, which inevitably leads to a loss of information. In this paper we demonstrate that  applying such a simple message passing scheme facilitates utilization of the full expressive power of hypergraphs. Contrary to most related work on hypergraph learning, here we respect the fact that hypergraph is a generalization of a graph and show that some popular graph-based approaches can be derived from our proposed HyperSAGE framework by selecting an appropriate maximum cardinality (cf. e.g. Section 3.3). We have also provided two important propositions on which a hypergraph based learning model should adhere to.
> The novelty of our proposed framework lies in (a) its generalizability, as multiple existing algorithms can be derived as a special case of HyperSAGE, (b) the flexibility it provides with two-level aggregation and sampling approach, and (c) the stability in the performance even while using less training data then the state of the art alternatives.
> ---
> *Q: In Equ. (3): N(v, e) -> |N(v_i, e)|, N(v_i) -> |N(v_i)|*
> R: Thank you very much for pointing this out. We carefully read the paper again, correcting this and several other typos that originally slipped our attention.
> ---
> *Q: it seems the reported performance of hypergraph-based methods is not better than the results of GAT, GCN. So, why should we use hypergraph-based methods?*
> R: Thank you for pointing to an important aspect requiring further clarification.  First, we would like to note that the real advantage of using hypergraphs over graphs is their ability to model complex heterogeneous datasets [1]. This has been shown in many fields and especially in multimedia [2,3,4,8]. For example, in the domain of visual object tracking, recently it has been shown that graphs cannot be used to model the temporal and spatial relations simultaneously [5]. In such multimodal problems, the use of hypergraphs as a more generic data representation is prefered or often the only option that does not involve oversimplifying assumptions.
> However, citation networks have been used as the most standard benchmark datasets in developing theoretical frameworks and showing the performance of geometric deep learning algorithms. For this reason,  most (if not all) related work ongraph/hypergraph based models published in venues such as NeurIPS and ICLR uses citation networks for evaluation. While our proposed HyperSAGE approach yields a competitive performance even on citation networks that can be naturally modelled with graphs, it is intended for use on much more heterogeneous datasets.
> In our follow up work, we have compared HyperSAGE with graph-based algorithms on the following two multimodal problems:
> 1.  Image classification on social multimedia network dataset taken from Flickr [6]. The results are convincing with an average improvement of 3.6% in terms of area under the ROC curve.
> 2. Autism Classification on Neuroimaging dataset (ABIDE 1 and ABIDE 2) [7]. Given the f-MRI data of subjects, the goal is to classify  them into autistic or non-autistic similar to [9]. The connections within each subject’s brain have been modelled by hypergraph and graphs for 750 different subjects. We found that our HyperSAGE method yields an accuracy of 66.1 ± 3.1 on this hypergraph classification task, whereas, a GCN based approach has a performance of  64.2± 4.1.
>
> If the reviewer deems it useful, we can include these additional experiments in the supplementary material to further show the importance and advantages of modelling data using hypergraphs.

---

> > ### Author Response · Authors · 2020-11-13
> > **References**
> >
> > [1] Wolf, Michael M., Alicia M. Klinvex, and Daniel M. Dunlavy. "Advantages to modeling relational data using hypergraphs versus graphs." 2016 IEEE High Performance Extreme Computing Conference (HPEC). IEEE, 2016.
> > [2] Arya, Devanshu, and Marcel Worring. "Exploiting relational information in social networks using geometric deep learning on hypergraphs." Proceedings of the 2018 ACM on International Conference on Multimedia Retrieval. 2018.
> > [3] Bu, Jiajun, et al. "Music recommendation by unified hypergraph: combining social media information and music content." Proceedings of the 18th ACM international conference on Multimedia. 2010.
> > [4] Kim, Eun-Sol, et al. "Hypergraph Attention Networks for Multimodal Learning." Proceedings of the IEEE/CVF Conference on Computer Vision and Pattern Recognition. 2020.
> > [5] Yan, Yichao, et al. "Learning Multi-Granular Hypergraphs for Video-Based Person Re-Identification." Proceedings of the IEEE/CVF Conference on Computer Vision and Pattern Recognition. 2020.
> > [6] McAuley, Julian, and Jure Leskovec. "Image labeling on a network: using social-network metadata for image classification." European conference on computer vision. Springer, Berlin, Heidelberg, 2012.
> > [7] Di Martino, Adriana, et al. "The autism brain imaging data exchange: towards a large-scale evaluation of the intrinsic brain architecture in autism." Molecular psychiatry 19.6 (2014): 659-667.
> > [8] Arya, Devanshu, Stevan Rudinac, and Marcel Worring. "HyperLearn: a distributed approach for representation learning in datasets with many modalities." Proceedings of the 27th ACM International Conference on Multimedia. 2019.
> > [9] Parisot, Sarah, et al. "Spectral graph convolutions for population-based disease prediction." International conference on medical image computing and computer-assisted intervention. Springer, Cham, 2017.

---

### Official Review · AnonReviewer5 · 2020-11-05
**This paper showed the limitations of existing graph neural networks on hyper graphs, and presented a novel inductive hypergraph learning framework to encode the intra-relations (within a hyperedge) as well as inter-relations (across hyperedges) from the hypergraph. The extensive experiments confirm its effectiveness and robustness.**

**Rating:** 6
**Confidence:** 4

**Review:**

Overall, this paper is well-written. However, the intuition of the proposed framework is not very clear, and the model efficiency needs to be analyzed and evaluated compared to other baselines.

This paper analyzed the key issues of the existing message-passing graph convolutional networks. That is, the multiply stacked layer might be over-fitting and over-smoothing. Thus it proposed to choose the neighbors from the entire graph based on the structure-aware and feature-aware relatedness rather than simply choosing the local neighborhood. However, the motivations of the proposed structure-aware and feature-aware teleport functions are not very convincing, and Table 3 shows that the performance improvement of TeleGCN might largely be induced by model architecture rather than the proposed TeleGCL.

Specifically, the pros and cons of this paper are summarized as follows.
Pros:
[1] It proposed a two-level message-passing strategy for neighborhood aggregation in the hypergraph.
[2] The proposed HyperSAGE framework is inductive.
[3] The experiments demonstrate its effectiveness and robustness compared to baselines.

Cons:
[1] It required the message aggregation scheme had the property of hypergraph isomorphic equivariance. Two question are that can the examples of Figure 2(b) be distinguished by the proposed scheme in Eq. (3), and can the proposed be guaranteed to distinguish the hypergraphs which not isomorphic?
[2] It is not clear what MLP+HLR is in Table 3?
[3] In Section 4.3, it is confusing why to split each of the datasets into a train-test ratio of 1:4. Figure 3 has shown that with such a train-test ratio, HyperSAGE might work just like the HyperGCN.
[4] The efficiency of HyperSAGE needs to be confirmed empirically compared to baselines as it argued that the condensed neighborhood could help improve the training efficiency.
[5] There is a typo in Eq. (3). Should it be |N(v_i, e)|/|N(v_i)|?

---

> ### Author Response · Authors · 2020-11-13
> **Response to Reviewer #1, Part 1/2**
>
> We would like to thank the reviewer for the constructive and valuable feedback that helped us improve the paper. Below we provide responses to the individual comments.
>
> ---
> *Q: However, the motivations of the proposed structure-aware and feature-aware teleport  .....*
> R: We believe there is an error on the side of the reviewer. The remark provided above seems to be completely unrelated to our method, and actually relates to another [ ICLR submission: ](https://openreview.net/forum?id=IpPQmzj4T_). However, pros and cons seem to be relevant to our paper. We will address the issues raised by the reviewer one by one.
> ---
> *Q: can the examples of Figure 2(b) be distinguished by the proposed scheme in Eq. (3)*
> R: We would like to point out that the paper does not have Fig. 2(b). However, the comment text  seems to be referring to Fig. 1(b) and we formulate our response with respect to it.
> The examples show in Fig. 1b can indeed be differentiated using our framework. While performing message passing using Eq. (3), our model can distinguish the two Fano planes as compared to the clique expansion-based methods. This is because HyperSAGE considers intra-edge neighborhood ($\mathcal{N}(v_i,\mathbf{e})$) and inter-edge/global neighborhood ($\mathcal{N}(v_i)$) as two different sets of vertices.
>  In this way, for instance vertex $v_2$ in Figure 1(b) has the same global neighborhood $\mathcal{N}(v_2) = \[ v_1,v_3,v_4,v_5,v_6, v_7 \]$ for both the Fano planes. However, its intra-edge neighborhood differs as in the first case v_2 for one of the edges ($\mathbf{e_k}$) has an intra-edge neighborhood $\mathcal{N}(v_2,\mathbf{e_k}) = \[v_1,v_6\]$ and for the other hypergraph $v_2$ for edge $\mathbf{e_m}$ has an intra-edge neighborhood $\mathcal{N}(v_2,\mathbf{e_m}) = \[v_1,v_4\]$. This difference is important to distinguish the two hypergraphs during message passing, as in one of them $v_2$ co-occurs with $v_1$ and $v_6$ while in the other it co-occurs with $v_1$ and $v_4$. Thus, the neighborhood feature aggregations would be different in the two scenarios, making HyperSAGE capable of distinguishing such hypergraphs.
> ---
> *Q: can the proposed be guaranteed to distinguish the hypergraphs which not isomorphic?*
> R: Yes, it can be guaranteed to distinguish the hypergraphs which are not isomorphic. Infact, when two hypergraphs are not isomorphic, a message passing neural network framework will be able to distinguish between them. The major issue with existing models is that they might generate different node embeddings for two hypergraphs which are isomorphic. We argue that HyperSAGE can solve such ambiguities by using Eq. 3. We hope to have clarified this point, but if the reviewer would prefer a more in-depth elaboration, we would be more than happy to provide it.
> ---
> *Q: It is not clear what MLP+HLR is in Table 3?*
> R: Thank you for pointing to an aspect requiring further clarification.  MLP+HLR refers to Multi-layer perceptron with hypergraph regularization. Multi-layer perceptron + explicit hypergraph Laplacian regularisation (MLP + HLR) regularises the MLP by training it with the loss given by $\mathcal{L} = \mathcal{L_0} + \lambda\mathcal{L_{reg}}$    and uses the hypergraph Laplacian for explicit Laplacian regularisation $\mathcal{L_{reg}}$. It is one of the most common baseline [Zhuo et al. '07]  used for node classification on hypergraphs. Details related to MLP+HLR will be added in the updated draft.
> ---
> *Q: In Section 4.3, it is confusing why to split each of the datasets  ...*
> R: Figure 3 summarizes an experiment with different train-test ratios, conducted to show that (a) HyperSAGE performs better than HyperGCN even with less training data, (b) the variance in accuracy of HyperSAGE is much lower as compared to HyperGCN, showing a more stable result and (c) since HyperSAGE uses all the neighborhood of a node (or a big sample), which is not the case with HyperGCN, for a similar level of performance HyperGCN needs more training samples. These three points prove our hypothesis about the advantages of using the entire relational information within a hypergraph over using only a subset (or a single) neighborhood or reducing a hypergraph to graph. Such approximation of neighborhood dents the performance of the model significantly and can only perform at par when provided with substantially more training data.
> With regard to the experiment presented in Section 4.3, in which we investigate different settings of parameter p in our proposed approach (cf. Table 3), a train-test ratio of 1:4 was chosen. Experimenting with the alternative train-test ratios yielded similar results to those presented in Table 3. Finally, we would like to remind the reviewer that HyperGCN is not an inductive approach and, therefore, cannot be applied to the problem addressed in Section 4.3.
> We hope that this answers the question. However, if a further clarification is needed, please let us know and we will provide a more in-depth response.

---

> > ### Author Response · Authors · 2020-11-13
> > **Response to Reviewer #1, Part 2/2**
> >
> > *Q: The efficiency of HyperSAGE needs to be confirmed empirically compared to baselines as it argued that the condensed neighborhood could help improve the training efficiency.*
> > R: Our proposed HyperSAGE model provides the user with a direct control over balancing the efficiency and accuracy (discriminative power). In terms of time complexity, given an attributed hypergraph $\mathcal{H}=(\mathcal{V, E})$ , with $\mathbf{X}$ as the feature matrix with each row representing node feature vectors.
> > Let $d$ be the number of initial features, $h$ be the number of hidden units, and $c$ be the number of labels. Further, let $T$ be the total number of epochs of training and let
> > $N =  \sum_{e\in\mathcal{E}} |e|$  ; $N_m =  \sum_{e\in\mathcal{E}} (2|e| -3)$ ;  $N_l = \sum_{e\in\mathcal{E}} |e|_{C_2}$
> >
> > then
> > * HyperSAGE takes $O(TN(1+h(d+c))$ time
> > * HyperGCN takes $O(T(N+N_mh(d+c)))$ time
> > * HGNN takes $O(TN_lh(d+c))$ time
> >
> > We will include these details in the supplementary material.
> >
> > *Q: There is a typo in Eq. (3). Should it be |N(v_i, e)|/|N(v_i)|?*
> > R: Thank you for pointing this out. We carefully read the paper again, correcting this and several other initially overlooked typos.

---

### Decision · Program_Chairs · 2021-01-07
**Final Decision**

**Decision:**

Reject

**Comment:**

The paper proposes a learning framework for Hypergraphs. The proposed method can be viewed as generalisation of GraphSAGE to hyper graphs. Though the paper emphasises that there is significant differences between Hypergraphs and Graphs and hence new methods are required. However, the proposed methods are not significantly different than that used for Graphs. Thus the novelty seems to be limited and hence it is difficult to strongly argue for acceptance.